# Follicular Fluid Components in Reduced Ovarian Reserve, Endometriosis, and Idiopathic Infertility

**DOI:** 10.3390/ijms24032589

**Published:** 2023-01-30

**Authors:** Giulia Collodel, Laura Gambera, Anita Stendardi, Fabiola Nerucci, Cinzia Signorini, Caterina Pisani, Marzia Marcheselli, Francesca Letizia Vellucci, Silvana Enrica Pizzasegale, Lucia Micheli, Elena Moretti

**Affiliations:** 1Department of Molecular and Developmental Medicine, University of Siena, 53100 Siena, Italy; 2Fertility Center, AGI Medica, 53100 Siena, Italy; 3Division of Clinical Pathology, University Teaching, Hospital of Siena, 53100 Siena, Italy; 4Department of Economics and Statistics, University of Siena, 53100 Siena, Italy; 5NBFC, National Biodiversity Future Center, 90133 Palermo, Italy; 6Department of Medicine, Surgery and Neurosciences, University of Siena, 53100 Siena, Italy

**Keywords:** endometriosis, female idiopathic infertility, follicular fluid molecules, inflammation, oxidative stress, lipid mediators, reduced ovarian reserve

## Abstract

Follicular fluid (FF) molecules, and their increase or decrease, can contribute to appropriate follicular growth and oocyte maturation, thus being related to female infertility conditions. In this paper, we studied the changes and the relationships of some biochemical components, hormones, antioxidant enzymes, F_2_-Isoprostanes (F_2_-IsoPs), and resolvin (Rv) D1 in the FF of infertile women with different reproductive conditions such as endometriosis, reduced ovarian reserve, and idiopathic infertility during assisted reproductive techniques (ART). In the whole population, positive correlations between albumin (ALB)/iron (Fe), ALB/beta-2-microglobulin (B2MG), and F_2_-IsoPs/RvD1 were detected in the FF. In FF from aged women, increased levels of follicle stimulating hormone (FSH) and reduced anti-Müllerian hormone (AMH) levels were associated with a worse oocyte quality. The negative ART outcome was influenced by patient age and AMH, B2MG, and FSH levels. Moreover, the reduced ovarian reserve condition was characterised by a significant decrease in oocyte number and quality, AMH amount, and lactate dehydrogenase (LDH) activity, as well as by an increase in age and FSH levels. In the presence of endometriosis, high levels of MDA and RvD1 were detected in FF, with a decrease in luteinising hormone (LH). Finally, among the molecules examined, none characterised the condition of idiopathic infertility. These data could support the identification of new FF markers in different reproductive disorders, suggesting the need for personalised therapeutic approaches and optimised ART outcomes. In particular, the evaluation of resolvins and lipid mediators in FF could be a promising field of investigation with which to understand the entity of oxidative stress and inflammation in some female infertility conditions.

## 1. Introduction

Intercellular communication allows the oocyte to determine its own fate by influencing the intrafollicular microenvironment, which in turn provides the necessary cellular functions for oocyte developmental competence, which is defined as the ability of the oocyte to complete meiosis and undergo fertilisation, embryogenesis, and term development [1]. 

The follicular fluid (FF) contains a complex mixture of steroids, metabolites, polysaccharides, proteins and small peptides, reactive oxygen species (ROS), and antioxidant enzymes [2]. Apart from acting directly as cellular signals, the mutual interaction of many of these FF molecules, or their increase or decrease, will probably also contribute to appropriate follicular growth and oocyte maturation. Many of the identified proteins relate to biological processes such as acute response signalling, inflammation, coagulation, and complement cascades [3]. 

Approximately one third of subfertility is unexplained; it is possible that a proportion of these cases may be due to an impact of the FF milieu on oocyte quality. Previous studies have identified several lifestyle and clinical factors which are associated with the composition of the FF. Lower FF antioxidant enzyme levels have been reported in older women compared to younger ones [4]. 

Women with either a reduced ovarian reserve or advanced maternal age have altered follicular cell metabolism, FF metabolites, and progesterone production [5]. Cytokines and immune cells, including interleukin (IL)-6, IL12, sHLA-G, macrophages, NK cells, and lymphocytes, might influence the oocyte–granulosa–cell complex, altering the balance of immune content; this process is possibly involved in infertility associated with immune-mediated diseases such as endometriosis [6].

Recently, it was shown that the FF of idiopathic infertile women has bio macromolecular differentiation using infrared spectra measurements and chemometric analysis [7].

Analysis of the proteomic composition of human FF has been previously proposed as a potential tool for oocyte quality evaluation in order to develop an efficient method to investigate the human FF proteome and peptidome components [8].

The study of metabolites may suggest personalised treatment in female infertility. Kermack et al. [9] stated that the fatty acid composition of human FF is altered by a 6-week dietary intervention that includes marine omega-3 fatty acids. Moreover, cells derived from FF exhibited stem cell characteristics, which could be useful for regenerative medicine applications and cell-based therapies [10].

Recent results indicated that the levels of F_2_ isoprostanes (F_2_-IsoPs, which are oxygenated products from arachidonic acid metabolism) and resolvins (lipid mediators of resolution of inflammation) have been promising biomarkers for the evaluation of FF quality [11].

In this paper, we dosed a series of molecules (biochemical components, hormones, antioxidant enzymes, F_2_-IsoPs and resolvin (Rv) D1 in the FF of selected women during assisted reproductive treatment (ART). Women were infertile due to endometriosis, reduced ovarian reserve, and idiopathic infertility. Correlations and comparisons were carried out to suggest new markers for pathologies associated with female infertility. 

## 2. Results

We examined the FF of 72 infertile women (with reduced ovarian reserve, endometriosis, or idiopathic infertility). The FF samples were obtained on the day of ART. Levels of molecules such as albumin (ALB), vitamin B12 (B12), beta-2-microglobulin (B2MG), iron (Fe), IL-6, malondialdehyde (MDA), ascorbic acid (AA), F_2_-IsoPs, and RvD1 and hormones such as luteinising hormone (LH), follicle stimulating hormone (FSH), and testosterone (TESTO) were dosed. Lactate dehydrogenase (LDH), catalase (CAT), total creatine kinase (CK), glutathione (GSH), glutathione peroxidase (GP), and glutathione reductase (GR) activities were assayed. We considered age, blood anti-Müllerian hormone (AMH) levels, and oocyte/mature oocyte production for each patient.

Initially, a correlation analysis was performed considering all the 72 infertile women. It was observed that age was significantly negatively correlated with AMH amount and positively correlated with FSH levels, an increase of which negatively influences the oocyte number and quality. Interestingly, we found significant positive correlations between ALB/Fe (*r* = 0.42, *p* < 0.001), ALB/B2MG (*r* = 0.57, *p* < 0.001), and F_2_-IsoPs/RvD1 levels (*r* = 0.28, *p* = 0.033). The positive correlations of ALB with Fe and B2MG were detected in all groups examined subsequently.

Then, the infertile women were divided into two groups according to their age (≥38 and <38 years). In the group of women over 38 years of age, positive correlations were detected for B2MG/GP (*r* = 0.55, *p* = 0.007), F_2_-IsoPs/GSH (*r* = 0.51, *p* = 0.003), F_2_-IsoPs/RvD1 (*r* = 0.43, *p* = 0.036), ALB/Fe (*r* = 0.47, *p* = 0.004), and ALB/B2MG (*r* = 0.37, *p* = 0.033). In the group of women <38 years-old, positive correlations were found for B2MG with Fe (*r* = 0.44, *p* = 0.007), LDH (*r* = 0.44, *p* = 0.007), F_2_-IsoPs (*r* = 0.36, *p* = 0.041), and LDH/Fe (*r* = 0.50, *p* = 0.002). Negative correlations were detected between B2MG/MDA (*r* = −0.43, *p* = 0.026), oocytes number/FSH (*r* = −0.39, *p* = 0.018), mature oocytes number/FSH (*r* = −0.34, *p* = 0.043), and AMH/RvD1 (*r* = −0.52, *p* = 0.048). 

Moreover, some summary statistics are reported in Table 1 for each variable in each group together with the *p*-values of the Wilcoxon test of medians equality. 

In particular, the AMH (*p* = 0.007) levels, number of oocytes (*p* = 0.003), and number of mature oocytes (*p* = 0.004) were significantly different, with larger medians in the group of women < 38 years; the levels of FSH (*p* = 0.0004) were significantly different, with larger medians in the group of women ≥38 years (Table 1).

Secondarily, correlations in the groups of infertile women with positive or negative outcomes for ART were also investigated. In the group with positive ART outcomes, 3 women had endometriosis and 12 had idiopathic infertility; in the group with negative ART outcomes, 29 patients had reduced ovarian reserve, 19 had endometriosis, and 9 were affected by idiopathic infertility. The correlations between the variables appeared very different in these two groups. In the positive ART outcome group, a relevant role was shown for the antioxidant component, with GP being significantly negatively correlated with the mature oocytes number (*r* = −0.76, *p* = 0.029), MDA (*r* = −0.69, *p* = 0.059), CAT (*r* = −0.71, *p* = 0.049), and RvD1 levels (*r* = −0.74, *p* = 0.03); RvD1 was positively correlated with CAT (*r* = 0.97, *p* < 0.001). In the negative ART outcome group, F_2_-IsoPs/RvD1 (*r* = 28, *p* = 0.056) was positively correlated.

Additionally, in this case, some summary statistics are reported in Table 2 for each variable in each group together with the *p*-values of the Wilcoxon test of medians equality. Significant differences of AMH (*p* = 0.021), B2MG (*p* = 0.021), number of oocytes (*p* = 0.013), number of mature oocytes (*p* = 0.0016), FSH (*p* = 0.0069), and age (*p* = 0.005) were detected. In particular, medians of AMH, B2MG, AA, number of oocytes, and number of mature oocytes are larger in the group associated with positive outcomes in ART, whereas, in the same group, medians of FSH, MDA, and age were smaller. 

Finally, all of the indices evaluated in the fluids from the three groups with different reproductive conditions (reduced ovarian reserve, endometriosis, and idiopathic infertility) were compared. 

The reduced ovarian reserve group was composed of 29 patients: 21 over 38 years and 0 positive ART outcomes. The endometriosis group had 22 patients: 14 over 38 years and 3 positive ART outcomes. The idiopathic infertility group included 21 patients: 1 over 38 years and 12 positive ART outcomes. 

From Table 3, it is at once apparent that the reduced ovarian reserve group clearly showed a significant difference in the number of oocytes and the number of mature oocytes (*p* < 0.001 for each comparison), LH (*p* < 0.05 with respect to the endometriosis group), FSH (*p* < 0.001 for each comparison), LDH (*p* < 0.05 for each comparison), MDA (*p* < 0.05 with respect to the endometriosis group), RvD1 (*p* < 0.1 considering idiopathic infertility group), AMH (*p* < 0.05 with respect to the endometriosis group and *p* < 0.001 with respect to the idiopathic infertility group), and age (*p* < 0.001 compared to the idiopathic infertility group and *p* < 0.05 with respect to the endometriosis group). 

Moreover, in the comparison between the endometriosis group and the idiopathic infertility group, there is a significant difference in age (*p* < 0.05), MDA (*p* < 0.05), and RvD1 (*p* < 0.1). 

Figure 1 shows the box plots allowing to conveniently visualise the differences among the three groups (endometriosis, idiopathic infertility, and reduced ovarian reserve) for the variables: age (a), AMH (b), number of oocytes (c), number of mature oocytes (d), LH (e), FSH (f), LDH (g), MDA (h), and RvD1( i). Reduced ovarian reserve group showed increased age and FSH levels and reduced AMH amount, oocyte quality, and LDH activity compared to other groups. In the endometriosis group, high MDA and RvD1 as well as low LH levels were detected. The idiopathic infertility group was not characterised by specific markers; low MDA and RvD1 levels were dosed.

## 3. Discussion

FF is an unquestionable player in female fertility, although many doubts remain in understanding its role in different pathological conditions. FF components influence oocyte quality and, consequently, female fertility; these components are affected by the age of the patient, the presence of inflammation and altered hormone levels.

In this paper, we studied the changes and the relationships of some biochemical components, hormones, antioxidant enzymes, F_2_-IsoPs, and RvD1 in FF of women with different reproductive conditions. 

A similar paper investigating the metabolic, enzymatic, and protein composition of FF in mares suggested that the nutritional environment of oocytes and follicular cells could improve the clinical diagnosis of infertility [12].

In our population, which was composed of patients affected by reduced ovarian reserve, endometriosis, and idiopathic infertility, positive correlations were found for ALB, Fe, and B2MG levels and LDH activity, indicating that the correct interactions of these molecules may contribute to maintain a balanced environment in FF. ALB represents a major component of both the serum and the FF, with a role in osmotic regulation and antioxidant properties. The redox status of ALB in FF has been evaluated, and the presence of a free highly reactive thiol group at position 34 (Cys-34) defines the reduced form; when the thiol group is bound to thiol-containing compounds, this is defined as the oxidised form [13]. The level of reduced ALB in FF has a positive reproductive impact, suggesting that FF ALB stands as an important means to buffer oxidative conditions. In fact, FF samples associated with viable oocytes exhibited significantly higher levels of reduced ALB compared to FF samples surrounding structurally degenerated oocytes, the levels of oxidised ALB of which were higher [14]. The role of ALB as a transporter is also evident from the positive correlations with Fe and B2MG. Gonzales et al. [15] reported that ALB complexes inorganic Fe in human bronchioalveolar fluid, enhancing the generation of ROS. 

B2MG is a protein present on the plasma membrane of nucleate cells. In a previous study to identify markers of IVF outcome, the follicular concentration of 47 proteins was correlated to oocyte cleavage. The group with cleavage at a significantly higher level was observed for six proteins: C3 complement fraction and ceruleoplasmin, alpha-1-antitrypsin and transferrin, and alpha 2-macroglobulin and beta 2-microglobulin [16]. Mihou et al. [17] reported a prognostic significance of multiple myeloma staging based on the combination of B2MG and ALB. Interestingly, the RvD1 amount was inversely associated with AMH levels in aged women. RvD1 is an endogenous lipid mediator derived from the enzymatic oxygenation of docosahexaenoic acid, and its increase may accelerate the resolution of inflammation involving different pathways [18,19]. AMH is clinically useful as a screening tool for diminished ovarian reserve [20]. Perturbations in serum AMH are linked to a variety of pathological conditions, for instance, polycystic ovary syndrome and adult granulosa cell tumours; and it can be used as a tumour marker to gauge response to therapy and monitor recurrence [21]. Our data indicate a relationship between AMH and age; the involvement does not seem relevant in endometriosis or idiopathic infertility. 

In men, the RvD1 amount was positively correlated with F_2_-IsoP level and reduced sperm quality [22]. F_2_-IsoPs may be considered a ‘gold standard’ biomarker of endogenous lipid peroxidation (LPO) [23]. Therefore, the positive correlations between F_2_-IsoPs/RvD1 detected both in the FF from women over 38 years or with negative ART outcomes seems to indicate that RvD1, along with other markers of oxidative stress (OS) and inflammation such as fatty acid contents, also increases in females. Moreover, in patients with positive ART outcomes, RvD1 was positively correlated with CAT and negatively with GP, probably showing an active environment in which to balance OS. 

In the case of polycystic ovary syndrome, a high pro-inflammatory mediator/specialised resolving mediator ratio was documented in serum [24]. These results agree with our data and suggest that a targeted supplementation with specialised resolving mediators or their precursors may be a valuable novel therapeutic strategy worth further investigation in the management of infertile women. 

Our study confirmed that the increased levels of FSH and reduced AMH in FF from aged women are associated with worse oocyte quality. MDA was increased but not significantly different in the group of patients divided for age or ART outcomes. MDA negatively correlated with GP and B2MG. MDA is a marker of LPO for which an increase is associated with the presence of OS. It has been reported that there is a decrease in systemic antioxidant capacity in advancing age, [25], which may impact the quality of eggs and embryos.

In males, CK catalyses the regeneration of ATP from the chemical shuttle between CK and CK phosphate, which is important in sperm function [26]. CK activity was increased in semen samples from subjects with reduced sperm vitality and pH [27] and poor sperm quality [28]. Lee et al. [29] suggested high CKB expression in the cumulus cells as a biomarker of good-quality embryos in both older and younger age groups. 

In our study, CK did not present significant differences, even if it resulted higher in the group of women over 38 years. 

Finally, when comparing the FF composition of patients grouped according to their pathology, we confirmed data reported in the literature, with the addition of new information. The reduced ovarian reserve condition was characterised by a significant decrease in oocyte number and quality, AMH amount, and LDH activity, as well as by an increase in age and FSH. Previously, FF LDH levels have shown an association with patient age and follicle size [30]. The role of age, FSH, and AMH levels in this condition is known [31].

Endometriosis was characterised by an increase in MDA and RvD1 levels, with a decrease in LH. Other authors proposed that endocrine and inflammatory factors such as LH and PGE_2_ impair ART outcomes in patients with endometriosis [32]. 

Many studies have suggested an association between endometriosis and ovarian oxidative imbalance [25,33,34]. In the present research, the involvement of OS in endometriosis is indicated by increased MDA levels, but the novelty of our data is represented by the involvement of increased RvD1 levels in this pathological condition. These results agree with those obtained in males. Increased levels of MDA, CK activity, F_2_-IsoPs, and RvD1 have been detected in the semen of patients affected by leukocytospermia and varicocele, suggesting an involvement of these molecules in inflammation and OS [22,27,35].

The role of other resolvins in FF has recently been investigated. In vivo, the predictive value of RvE1 for oocyte quality was detected; moreover, in vitro experiments revealed the cellular mechanism of RvE1 in improving oocyte quality by decreasing the cumulus cell apoptotic rate [36]. 

Among the molecules examined, no marker for idiopathic infertility was found. This is not surprising because the group includes many different conditions that are unknown as yet. Low levels of RvD1 and F_2_-IsoPs could exclude the presence of inflammation in endometriosis and, to a lesser extent, in reduced ovarian reserve.

## 4. Conclusions

Oocyte competence and embryo development are both influenced by multiple factors present in the follicular environment. The analysis of FF components which are available during oocyte retrieval provides information on metabolic changes in this microenvironment [37]. In particular, the evaluation of resolvins and lipid mediators in FF could be a promising field of investigation with which to understand the entity of OS and inflammation in some female infertility conditions and to optimise ART outcomes. Because of the association between metabolism and inflammation, a panel of markers should be explored simultaneously in a larger group of patients in order to identify new FF markers in the various reproductive disorders, suggesting personalised therapeutic approaches.

## 5. Materials and Methods

### 5.1. Patients

This study was conducted at the AGI Medica Fertility Center (Viale Toselli 94/F, Siena, Italy). During the study period (2020–2022), 72 consecutive infertile women were selected and included (aged 32–42). They were affected by reduced ovarian reserve, endometriosis, or idiopathic infertility.

In the first group, we included 29 women with poor ovarian response (aged 32–42). This usually indicates a reduction in follicular response, resulting in a smaller number of retrieved oocytes [38].

The second group was composed of 22 women affected by endometriosis (aged 35–42); a chronic, inflammatory disease defined as the presence of endometrium-like tissue outside the uterus [39]. The most frequent symptoms were dysmenorrhea, deep dyspareunia, dysuria, dyschezia, painful rectal bleeding or haematuria, fatigue, and infertility. Due to the different opinions in the diagnosis of this disease, we included only women with a previous histological diagnosis of endometriosis in this group [40].

In the third group, we included 21 females with unexplained infertility (aged 32–40). This diagnosis was made after excluding common causes of infertility using standard fertility investigations, including the assessment of ovulation and tubal patency test. 

Firstly, the infertile women were divided into two groups according to their age: <38 years (36 patients) and ≥38 years (36 patients). Among women aged ≥38 years, 21 had reduced ovarian reserve, 14 had endometriosis, and 1 had idiopathic infertility. In the group of patients aged <38 years, 8 had reduced ovarian reserve, 8 had endometriosis, and 20 had idiopathic infertility.

Secondarily, the infertile women were placed into groups for positive or negative outcomes for ART.

All of the patients respected these inclusion selection criteria: normal karyotype, BMI < 25 kg/m^2^, and the absence of infections. We did not include carrier patients with chronic diseases and those receiving radiotherapy, chemotherapy, or medication. Selected women did not take an oral antioxidant supplement for at least 3 months before the study and did not have a history of recreational drug use and alcohol consumption. Women were non-smokers. All of them were nulliparous when starting ovarian stimulation.

Ovarian hyperstimulation was performed attending short antagonist protocol. Menstrual cycle is normally induced through the oestro–progestin pill, and the patients received recombinant gonadotropins on day II or day III of the menstrual cycle after ultrasound examination. Injections of GnRH antagonists were administered flexibly (0.25 mg per day if the largest follicle was 14 mm and with oestrogen levels >300 pg/mL). The dose of gonadotropins was adjusted according to ovarian response, as detected by ultrasound examination and hormonal profile. Ovulation triggering was achieved by human chorionic gonadotropin (HCG) injection when follicles of >16 mm were present in the ovaries. At 34–36 h after the intramuscular injection of HCG, FF was retrieved using transvaginal ultrasound-guided aspiration. The oocyte aspiration needle is washed with 5 mL of culture medium (Cook, Follicle Flush Buffer) before and after aspiration. 

After the oocytes were found in the FF and placed in the appropriate culture dishes, the remaining FF was centrifuged. To avoid contamination by the blood or the flushing medium during oocyte retrieval, only the supernatant free of blood contamination after centrifugation at 3000 rpm for 10 min was extracted. For the analysis, 2 mL of centrifuged FF was taken and stored in 2-mL cryotubes at −80 °C until use.

When signing the informed consent to the treatment, patients who had been referred to our Clinic for IVF treatment accepted or did not accept the possibility that the FF not used for the treatment can be used for scientific research purposes. Patients who accepted this possibility and who met the inclusion criteria were enrolled in our study. The internal institutional ethics committee approved this study. Informed written consent to participate at the research protocol was obtained from all patients.

### 5.2. Clinical Biochemistry Determinations

The stored FF samples were then thawed at room temperature. 

FF aliquots were tested using a COBAS 8000 modular analyser (Roche Diagnostics, Mannheim, GmbH, Germany) by means of two analytical modules: C702, the high-throughput clinical chemistry module, and E602, the immunoassay module.

We measured the following parameters in FF: iron (Fe, µg/dL), albumin (ALB, g/dL), beta-2-microglobulin (B2MG, mg/L), creatine kinase (CK, UI/L), lactate dehydrogenase (LDH, U/L), testosterone (TESTO, ng/mL), luteinising hormone (LH, mUI/mL), follicle stimulating hormone (FSH, mUI/mL), vitamin B12 (B12, pg/mL), and interleukin-6 (IL-6, pg/mL).

For the analytes measured in module C702 (Fe, ALB, CK, LDH), COBAS 8000 calibration was performed with the human lyophilised serum calibration Cfas (Roche Diagnostics, Mannheim, GmbH, Germany). The Cfas (calibrator for automated systems) is a universal calibrator for adjusting most photometric methods. B2MG calibration was performed with a specific calibrator (Calibrator B2MG, Roche Diagnostics, Mannheim, GmbH, Germany).

Human lyophilised serum PreciControl Clin Chem level 1 was used as a normal control, and PreciControl Clin Chem level 2 was used as a pathological control (Roche Diagnostics, Mannheim, GmbH, Germany).

The B2MG (Roche Diagnostics, Mannheim, GmbH, Germany) level 1 and level 2 control set was used as quality control.

For the analyte measured in module E602 (TESTO, LH, FSH, B12, and IL-6), we used a specific calibrator for each analyte (Roche Diagnostics, Mannheim, GmbH, Germany). PreciControl Universal levels 1 and 2, PreciControl Varia levels 1 and 2, and PreciControl Multimarker levels 1 and 2 (Roche Diagnostics, Mannheim, GmbH, Germany) were used as normal and pathological controls, respectively.

### 5.3. F_2_-Isoprostane (F_2_-IsoP) Determination

In FF samples, F_2_-IsoP amounts were measured by a mass spectrometry analysis. After a sample purification step by solid phase extraction and a derivation procedure, the ion measured in GC/NICI-MS/MS analysis was 15-F_2t_ -IsoPs (one of the most represented isomers of F_2_-IsoPs and known as 8-epi-PGF_2a_). The detected product ion was at *m*/*z* 299 (precursors ions was at *m*/*z* 569) [a47] [41]. For each sample, the detected F_2_-IsoPs was referred to as the calibration curve (Cayman Chemical, Item No. 16350). F_2_-IsoP levels were reported as pg/mL.

### 5.4. Resolvin (Rv) D1 Assay

Analysis of RvD1 in FF was carried out by a quantitative sandwich enzyme-linked immunosorbent assay (ELISA) (MBS2601295-96 Rabbit resolving (Rv) D1 NR 1 MyBioSource). Spectrometric analysis was performed at 450 nm, and RvD1 amounts were determined by making reference to the standard curve ranging from 2000 pg/mL to 31.2 pg/mL [22].

### 5.5. Total Glutathione (GSH) 

An aliquot of FF was added to an equal volume of 10% metaphosphoric acid and promptly centrifuged at 2000× *g* for 10 min at 0 °C with a Thermo IEC CENTRA CL3R refrigerated centrifuge. Next, the supernatant was taken and immediately stored at –80 °C until use. A micro-assay procedure was utilised for measuring the GSH levels in the supernatant [42]. The method uses Ellman’s reagent (DTNB) for the derivatisation of thiol groups and photometric quantification of a released nitro-chromophore based on an enzymatic method with the readings at 415 nm. Results were expressed in nmol/mL.

### 5.6. Ascorbic Acid (AA) Assay

The levels of ascorbic acid (AA) were measured in the aliquot of FF acidified with 10% metaphosphoric acid by an HPLC method [43], with minor modifications. Initially, the supernatants were filtered (Anotop 0.2 μm, Merck), and 20 μL was injected into a high-performance liquid chromatography (HPLC) column. 

A reverse-phase HPLC method with UV detection was used to quantify AA. The HPLC-UV system (Waters 600 E System Controller, Milford, MA, USA) was equipped with a Waters Dual λ 2487 UV detector (Milford, MA, USA) set at 262 nm.

Ultrasphere ODS (C18) 5 µm HPLC Columns (Beckman, San Ramon, CA, USA) were used with acetonitrile–water (49/51, *v*/*v*) as the mobile phase at a flow rate of 0.8 mL/min. The concentrations of AA were calculated by peak areas determined using an Agilent 3395 integrator (Agilent Technologies, Santa Clara, CA, USA) and were expressed as nmol/mL.

### 5.7. Malondialdehyde (MDA) Assessment

To prevent artifact oxidations of polyunsaturated free fatty acids, immediately at the time of collection, an aliquot of FF (at least 0.2 mL) was added to an equal volume of tris-HCl 0.04 M and acetonitrile containing 0.1% butylated hydroxytoluene. This was followed by centrifugation at 2000× *g* for 10 min at 0 °C, and the supernatant was frozen at −80 °C until use.

The LPO in FF was estimated by calculating the malondialdehyde (MDA) levels according to the method of Shara et al. [44], with minor modifications. 

At the time of analysis, the supernatant was derivatised using 2,4-dinitrophenylhydrazine and immediately agitated and extracted with 5 mL of pentane; finally, the samples were dried using nitrogen. The calibration curve was derived using MDA as standard (range of concentrations from 0.5 nmol/mL to 10 nmol/mL).

An isocratic HPLC Waters 600 E System Controller HPLC (Milford, MA, USA) equipped with a Waters Dual λ 2487 UV detector (Milford, MA, USA) set at 307 nm was used to quantify the MDA hydrazone.

A 5 µm Ultrasphere ODS column C18 (Beckman, San Ramon, CA, USA) was used to separate the hydrazone derivative at the flow rate of 0.8 mL/min with a mobile phase consisting of acetonitrile:0.01 N HCl (45:55% *v*/*v*).

The MDA concentrations (nmol/mL) were calculated by peak areas determined using an Agilent 3395 integrator (Agilent Technologies, Santa Clara, CA, USA).

### 5.8. Catalase (CAT) Activity

To estimate catalase (CAT) activity, a microassay procedure was used [45]. An aliquot of FF was centrifuged at 4000× *g* for 15 min at 4 °C, and the supernatants were frozen at −80 °C until use. 

This method is based on the reaction of the enzyme CAT with methanol in the presence of an optimal concentration of hydrogen peroxide. The formaldehyde production was measured spectrophotometrically at 540 nm with 4-amino-3-hydrazino-5-mercapto-1,2,4-triazole (Purpald) as a chromogen. One unit of CAT activity is defined as the amount of enzyme that will cause the formation of 1 nmol of formaldehyde per minute at 25 °C. The results were expressed as U/mL.

### 5.9. Glutathione Reductase (GR) Activity

For the evaluation of glutathione reductase (GR) activity, an aliquot of FF was diluted (1:1) in cold 0.25 M sucrose (in 0.1 M phosphate buffer, pH 7.4) and centrifuged at 40,000× *g* for 20 min at 4 °C. The supernatants were stored at −80 °C until use.

The method is based on the increase in absorbance at 415 nm when 5,5′-dithiobis(2-nitrobenzoic acid) is reduced by the glutathione generated from an excess of oxidised glutathione in the presence of NADPH [46].

Samples were prepared in 96-well plates and absorbance was measured every 30 s for 3 min with a programmable microplate reader. The rate of increase in absorbance was directly proportional to the amount of enzyme GR in the sample. The results were expressed as U/mL.

### 5.10. Glutathione Peroxidase (GP) Assay

The FF was treated using the same procedure as described for GR. The glutathione peroxidase (GP) activity is quantitated by measuring the change in absorbance at 340 nm caused by the oxidation of NADPH [47]. One unit of GP activity is defined as the amount of enzyme that oxidises 1 µmol of NADPH at 37 °C per minute. Enzyme activity was expressed as U/mL. 

### 5.11. Statistical Analysis

Data were analysed using R version 4.1.1. The Pearson correlation coefficient was calculated to quantify the correlation between variables for all enrolled patients and, separately, for each group defined as patients aged <38 years, patients aged ≥38 years, patients with positive ART outcomes, and patients with negative ART outcomes. For each variable, the Shapiro–Wilk test was preliminarily performed to assess the normality distribution, and then the significance of the correlation coefficient was tested for any pair of variables for which the normality hypothesis had been accepted. Moreover, for the same pairs of variables, the values of the correlation coefficient are reported in the Appendix A together with the corresponding scatterplots. It is at once apparent that the number of variables for which the normality assumption had been accepted, referring to all enrolled patients and, separately, for each group defined in terms of age or ART outcomes, was very limited. Moreover, for some variables, the number of patients in the groups was not sufficiently large to implement parametric tests relying on asymptotic normal distributions. Therefore, to perform the comparison of two groups (patients aged <38 years vs. patients aged ≥38 years; patients with positive ART outcomes vs patients with negative ART outcomes) or three groups (according to the different reproductive conditions), the use of nonparametric tests seemed to be advisable. Comparisons between the two groups of patients formed considering age, as well as comparisons between the two groups of patients with ART positive or negative outcomes, were performed using the Wilcoxon test to evaluate the null hypothesis of equality of medians in the compared groups. A *p*-value less than 0.05 is typically considered to be statistically significant, in which case the null hypothesis should be rejected. 

Finally, patients were partitioned into three groups according to the pathology (endometriosis, idiopathic infertility, and reduced ovarian reserve), and differences between the groups were evaluated by using the Kruskal–Wallis test followed, only for significant cases, by Dunn’s test for multiple comparisons. 

## Figures and Tables

**Figure 1 ijms-24-02589-f001:**
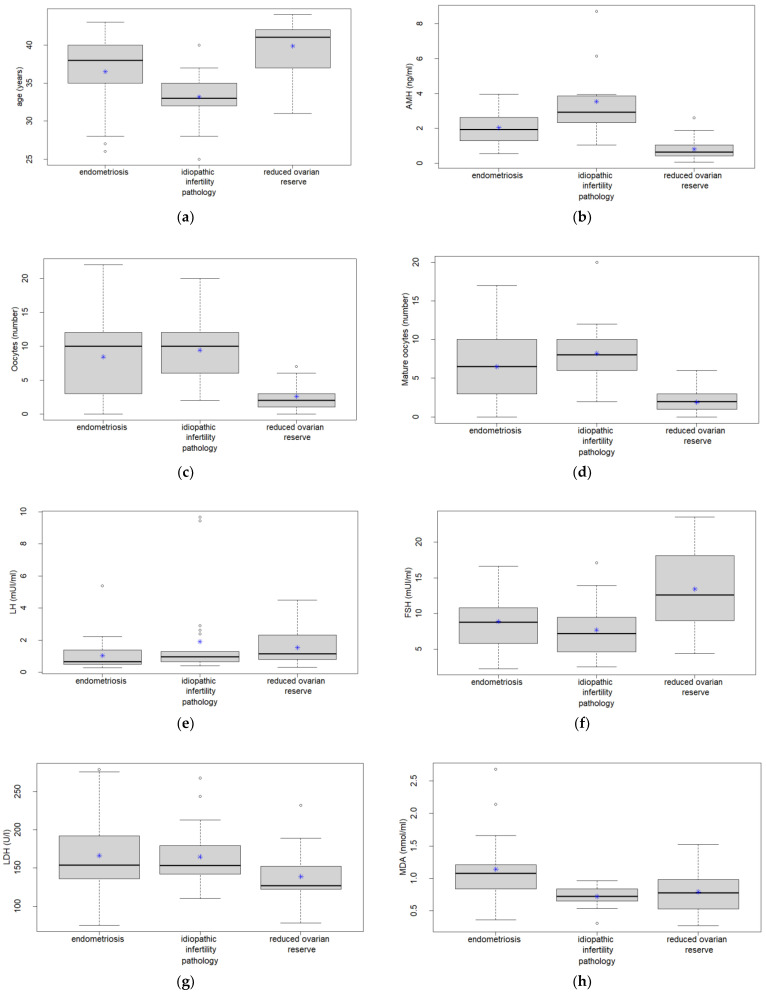
Box plots of the three groups (endometriosis, idiopathic infertility, and reduced ovarian reserve) for the variables: age (**a**), anti-Müllerian hormone (AMH, (**b**)), number of oocytes (**c**), number of mature oocytes (**d**), luteinising hormone (LH, (**e**)), follicle stimulating hormone (FSH, (**f**)), lactate dehydrogenase (LDH, (**g**)), malondialdehyde (MDA, (**h**)), and resolvins (RvD1, (**i**)) in terms of medians (bold lines), 25° and 75° centile (horizontal lines of the box), interquartile range (height of the box), spread outside 25° and 75° centile (dotted vertical lines), and outliers (dots). Stars denote mean values.

**Table 1 ijms-24-02589-t001:** Medians and 25°–75° centiles (in parenthesis) of albumin (ALB), vitamin B12 (B12), anti-Müllerian hormone (AMH), number of oocytes, number of mature oocytes, beta-2-microglobulin (B2MG), luteinising hormone (LH), follicle stimulating hormone (FSH), testosterone (TESTO), iron (Fe), lactate dehydrogenase (LDH), total creatine kinase (CK), malondialdehyde (MDA), glutathione (GSH), glutathione peroxidase (GP), glutathione reductase (GR), catalase (CAT), ascorbic acid (AA), F_2_-isoprostanes (F_2_-IsoPs), resolvin (Rv)D1, and Interleukin-6 (IL-6) assayed in follicular fluids of 72 infertile women divided into 2 groups according to their age (≥38 years, <38 years). In the last row, for each variable, the *p*-value of the Wilcoxon test. When *p*-value is greater than 0.05, ns, which stands for not significant, is reported.

	ALB	B12	AMH	Oocytes n°	Mature Oocytes n°	B2MG	LH	FSH	TESTO	Fe	LDH	CK	MDA	GSH	GP	GR	CAT	AA	F_2_-IsoPs	RvD1	IL-6
≥38 years	3.5 (3.1–3.8)	216.5 (171–289.8)	0.77 (0.6–1.8)	3 (1–7.2)	2.5 (1–5.2)	1.9 (1.7–2.1)	0.9 (0.6–1.6)	10.6 (8.7–15.9)	5.8 (3.1–10.4)	61.2 (48.4–75.1)	136.5 (122.8–160.5)	57.5 (40.2–76.2)	0.9 (0.6–1.1)	0.03 (0.02–0.05)	60.5 (51.9–70.6)	13.8 (11.5–15.9)	6.3 (4.6–8.3)	7.6 (3.8–12.2)	54.8 (45.5–67.9)	563 (413–764)	12.1 (8.5–16.9)
<38 years	3.4 (2.9–3.9)	239.5 (198–294.8)	2.3 (1.2–3.5)	8.5 (3.7–12)	6 (2.7–10)	2.1 (1.8–2.4)	0.8 (0.6–1.4)	7.7 (4.6–10.8)	4.3 (3–8.8)	69.4 (57.3–75.8)	152 (134–181.5)	45 (35–55.2)	0.6 (0.6–0.7)	0.05 (0.03–0.07)	77.8 (59.1–114.7)	12.4 (9–16.2)	5.6 (4.2–7.8)	10.5 (6.3–14.6)	54.1 (45.2–72.2)	459.4 (367.7–555.6)	12 (8–14.5)
*p*-values	ns	ns	0.007	0.003	0.004	ns	ns	0.0004	ns	ns	ns	ns	ns	ns	ns	ns	ns	ns	ns	ns	ns

**Table 2 ijms-24-02589-t002:** Medians and 25°–75° centiles (in parenthesis) of age, albumin (ALB), vitamin B12 (B12), anti-Müllerian hormone (AMH), number of oocytes, number of mature oocytes, beta-2-microglobulin (B2MG), luteinising hormone (LH), follicle stimulating hormone (FSH), testosterone (TESTO), iron (Fe), lactate dehydrogenase (LDH), total creatine kinase (CK), malondialdehyde (MDA), glutathione (GSH), glutathione peroxidase (GP), glutathione reductase (GR), catalase (CAT), ascorbic acid (AA), F_2_-isoprostanes (F_2_-IsoPs), resolvin (Rv)D1, and Interleukin-6 (IL-6) assayed in follicular fluids of 72 infertile women divided into 2 groups according to the assisted reproduction techniques (ART) outcome (negative, positive). In the last row, for each variable, the *p*-value of the Wilcoxon test. When *p*-value is greater than 0.05, ns, which stands for not significant, is reported.

	Age	ALB	B12	AMH	Oocytes n°	Mature Oocytes n°	B2MG	LH	FSH	TESTO	Fe	LDH	CK	MDA	GSH	GP	GR	CAT	AA	F_2_-IsoPs	RvD1	IL-6
Negative ART outcome	38 (35–42)	3.5 (2.8–3.8)	237 (182–310)	1 (0.6–2.2)	4 (1–10)	3 (1–7)	1.9 (1.7–2.2)	0.9 (0.6–1.7)	10.3 (8.2–13.5)	6.1 (2.9–10)	62.8 (48–75.6)	147 (124–175)	47 (36–66)	0.9 (0.6–1.1)	0.03 (0.02–0.05)	60.5 (51.9–70.6)	13.8 (11.5–15.9)	6.3 (4.6–8.3)	7.6 (3.8–12.2)	54.8 (45.5–67.9)	563 (413–764)	12.1 (8.5–16.9)
Positive ART outcome	34 (32–37)	3.5 (3.4–3.8)	229 (205–271.5)	2.8 (2.4–3.7)	10 (5.5–11.5)	8 (5.5–11)	2.3 (1.9–2.4)	0.7 (0.6–1.2)	7.2 (4.6–9.1)	3.9 (3.2–5.7)	69.4 (63.2–78.4)	152 (138–175.5)	46 (43–56.5)	0.6 (0.6–0.7)	0.05 (0.03–0.07)	77.8 (59.1–114.7)	12.4 (9–16.2)	5.6 (4.2–7.8)	10.5 (6.3–14.6)	54.1 (45.2–72.2)	459.4 (367.7–555.6)	12 (8–14.5)
*p*-values	0.005	ns	ns	0.021	0.013	0.0016	0.021	ns	0.0069	ns	ns	ns	ns	ns	ns	ns	ns	ns	ns	ns	ns	ns

**Table 3 ijms-24-02589-t003:** *p*-values for multiple and pairwise comparison between the three groups with different reproductive conditions. Only variables giving rise to *p*-values less than 0.05, reported in the last row, for the multiple comparison Kruskal-Wallis test are listed in the first row. In the first row, for each variable, the sample size is reported and, in the second row, the number of patients in the endometriosis, idiopathic infertility and reduced ovarian reserve group, respectively. In the inner cells, *p*-values of the Dunn test for pairwise comparisons. Antimullerian hormone (AMH), oocytes number, mature oocytes number, luteinising hormone (LH), follicle stimulating hormone (FSH), lactate dehydrogenase (LDH), malondialdehyde (MDA), resolvins (RvD1).

	Age	AMH	Oocytes n°	Mature Oocytes n°	LH	FSH	LDH	MDA	RvD1
sample size	62	38	62	62	62	62	62	50	55
n° patients for group	22, 21, 29	7, 12, 19	22, 21, 29	22, 21, 29	22, 21, 29	22, 21, 29	22, 21, 29	16, 13, 21	18, 18, 19
endometriosis-idiopathic infertility	0.011	0.147	0.375	0.162	0.181	0.317	0.914	0.040	0.074
endometriosis-reduced ovarian reserve	0.022	0.054	<0.001	<0.001	0.039	0.011	0.035	0.036	0.801
idiopathic infertility- reduced ovarian reserve	<0.001	<0.001	<0.001	<0.001	0.503	<0.001	0.041	0.692	0.080
*p*-value	<0.001	<0.001	<0.001	<0.001	0.042	<0.001	0.017	0.021	0.047

## Data Availability

The data used to support the findings of this study are available from the corresponding author upon request.

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
