# Peer review of "Follicular Fluid Components in Reduced Ovarian Reserve, Endometriosis, and Idiopathic Infertility"

_ijms, 2023, doi:10.3390/ijms24032589_

Round 1

Reviewer 1 Report

This study by Collodel et al. aimed to examine the composition of follicular fluid of infertile women with different diagnoses (reduced ovarian reserve, endometriosis, and idiopathic infertility). They also evaluated the possible statistical correlations between these components and the outcomes of artificial reproductive techniques.

This work investigates important aspects of reproductive medicine and has clinical significance as well, although many aspects of the study are not clear, and questions have arisen that need to be answered by the authors.

Format, grammar:

Above all, the language of the manuscript needs to be improved. Authors should seek professional help from an expert.

·         In the Introduction, lines: 18-19, the size and style of half of the sentence are different.

·         line 335: beta 2 microglobulin is already abbreviated B2MG, therefore the short form should be used throughout the text  

·         Ascorbic acid has been also abbreviated; therefore authors should use this form instead of full name, same problem with catalase also

·         line 420: use quantified instead of quantitied

·         The resolution of Table 3 is poor. The labeling should be with bigger fonts. The applied statistical comparisons both on the graph (with asterisk) and in the legend are missing.

·         The text is not edited correctly. Between lines 266-281 the authors left the text of the template in the manuscript.

Comments on content

·         I could not find any ethics number in the manuscript. This is very important to correct.

·         Materials and methods: the age, parity and gravity data of the women included in the study should be provided

·         line 101-105: the groups and patients characteristics data are instead belonging to the materials and methods section.

·         Authors should provide the R values of all comparisons in the case of correlation analysis and the scatterplots displaying the relationship between the variables should be presented at least as supplementary data.

·         The figure legends of Table 1 and 2 authors should provide information about the applied statistical analysis. Therefore, the asterisks are unnecessary if they are not displayed in the table.

Author Response

This study by Collodel et al. aimed to examine the composition of follicular fluid of infertile women with different diagnoses (reduced ovarian reserve, endometriosis, and idiopathic infertility). They also evaluated the possible statistical correlations between these components and the outcomes of artificial reproductive techniques.

This work investigates important aspects of reproductive medicine and has clinical significance as well, although many aspects of the study are not clear, and questions have arisen that need to be answered by the authors.

Format, grammar:

Above all, the language of the manuscript needs to be improved. Authors should seek professional help from an expert.

The paper has been revised by a specialized service. We submitted a certificate.

  • In the Introduction, lines: 18-19, the size and style of half of the sentence are different.

We corrected as rightly required

  • line 335: beta 2 microglobulin is already abbreviated B2MG, therefore the short form should be used throughout the text 

We corrected as required. 

  • Ascorbic acid has been also abbreviated; therefore authors should use this form instead of full name, same problem with catalase also

      We corrected as required

  • line 420: use quantified instead of quantitied

We corrected as required

  • The resolution of Table 3 is poor. The labeling should be with bigger fonts. The applied statistical comparisons both on the graph (with asterisk) and in the legend are missing.

We completely agree with your comments. In the revised manuscript Table 3 is Figure 1 whose caption has been changed according to your suggestion. Moreover, we emphasized that Figure 1 allows only a descriptive comparison between groups.

  • The text is not edited correctly. Between lines 266-281 the authors left the text of the template in the manuscript.

We are sorry, we corrected.

Comments on content

  • I could not find any ethics number in the manuscript. This is very important to correct.

      We better explained in the text the protocol of Fertility Center, AGI Medica.

  • Materials and methods: the age, parity and gravity data of the women included in the study should be provided

We inserted the information

  • line 101-105: the groups and patients characteristics data are instead belonging to the materials and methods section.

      We moved the sentence in the Materials and methods section.

  • Authors should provide the R values of all comparisons in the case of correlation analysis and the scatterplots displaying the relationship between the variables should be presented at least as supplementary data.

Pearson correlation coefficients have been reported in the revised manuscript. Scatter plots of pairs of variables giving rise to Pearson correlation coefficient significantly different from zero are reported in the Supplementary file.

  • The figure legends of Table 1 and 2 authors should provide information about the applied statistical analysis. Therefore, the asterisks are unnecessary if they are not displayed in the table.

The captions of Table 1 and Table 2 have been modified according to your suggestion.

Reviewer 2 Report

The authors tried to find correlations between some follicular fluid compounds and different forms of women infertility. The study is interesting but there are some methodological and statistical issues that need to be resolved. 

- line 73-78: this is not needed in introduction

-if you write in tables that p values under 0.05 were recognized as significant, than p value of exact 0.05 isn't significant if we are being strict, for instance in line 98, and also in tables 1 and 5 for MDA (also in line 130 and 135)

- line 108, p value 0.07?

-lines 266-281  (you forgot to delete this text)

-ovarian hyperstimulation protocol should be described more precisely

- line 317: only supernatant was stored or whole sample?

- Statistics: Pearson correlation is used to analyze parametric data but you presented you data as non-parametric (mediana with IQR). Why? All other tests you performed were non-parametric? 

- did medical ethics committee approved the study?

- folicular fluid can sometimes be very bloody. How did you deal with this? Because contamination with blood could have major impact on results. Also, the oocyte aspiration needle is usually washed with culture medium (flushing or MOPS or something similar) before and after aspiration. Did you consider this? Explain this in material and methods section. 

Author Response

Referee 2

The authors tried to find correlations between some follicular fluid compounds and different forms of women infertility. The study is interesting but there are some methodological and statistical issues that need to be resolved. 

- line 73-78: this is not needed in introduction

We deleted as required

-if you write in tables that p values under 0.05 were recognized as significant, than p value of exact 0.05 isn't significant if we are being strict, for instance in line 98, and also in tables 1 and 5 for MDA (also in line 130 and 135)

- line 108, p value 0.07?

We revised the statistical analysis.

-lines 266-281  (you forgot to delete this text)

We are sorry for the error.

-ovarian hyperstimulation protocol should be described more precisely

We added an accurate description.

- line 317: only supernatant was stored or whole sample?

We better described as rightly required.

- Statistics: Pearson correlation is used to analyze parametric data but you presented you data as non-parametric (mediana with IQR). Why? All other tests you performed were non-parametric? 

We considered Pearson correlation coefficient as it is the most widely adopted index for evaluating the linear relationship between data referring to two quantitative variables. Spearman correlation coefficient could have been alternatively computed exploiting only the ordering induced by the values of the variables, but not the values themselves, thus obtaining information only about the existence of a monotonic relationship between the two variables. However, inference on the Pearson correlation coefficient relies on the assumption that the variables are normally distributed. Therefore, tests on the significance of the Pearson correlation coefficient have been performed only for those pairs of variables for which the normality hypothesis had been accepted by means of the Shapiro Wilk test.

Since one of the main goals of the work was the comparison of two groups (age greater than or equal to 38 vs age less than 38, positive vs negative ART outcome) or three groups (according to different reproductive conditions), parametric tests, which require the normality distribution of the considered variable, could have been performed just on a limited number of variables. To overcome this problem, the Wilcoxon, Kruskal-Wallis and Dunn nonparametric tests, essentially based on ranks, have been adopted and, coherently, in Tables 1 and 2 the three quartiles are reported. 

- did medical ethics committee approved the study?

We described the protocol of AGI Fertility Center

- folicular fluid can sometimes be very bloody. How did you deal with this? Because contamination with blood could have major impact on results. Also, the oocyte aspiration needle is usually washed with culture medium (flushing or MOPS or something similar) before and after aspiration. Did you consider this? Explain this in material and methods section. 

We deeply described the method.

Round 2

Reviewer 1 Report

The authors significantly revised the manuscript according to the suggestions. Still, 2 points require further clarification.

Table 3: It is not clear what is seen in the table. In the legend, the authors only explain the statistical analysis, but an informative title is also necessary. Moreover, the units of different rows and the sample size should also be indicated in the first line.

 Figure 1: The explanation of the results depicted in Figure 1 is missing from the text. Also, the unit of measured parameters is not presented in the y-axis of the diagrams. In the text, the authors should refer to Figure 1.

Author Response

Table 3: It is not clear what is seen in the table. In the legend, the authors only explain the statistical analysis, but an informative title is also necessary. Moreover, the units of different rows and the sample size should also be indicated in the first line.

We included a title in Table 3. Moreover, in the first row, for each considered variable, the sample size and the number of patients in each group for which the value of the variable is available is now reported.

Figure 1: The explanation of the results depicted in Figure 1 is missing from the text. Also, the unit of measured parameters is not presented in the y-axis of the diagrams. In the text, the authors should refer to Figure 1.

We reported the unit of measurements in the y-axis of the boxplots in Figure 1 and we added a description in the text.

Reviewer 2 Report

The authors improved the significantly the manuscript but there is still some issues with data presentation and statistics. You wrote that some data were normally distributed and therefore the pearson correlation test was used but you presented all data as medians? If some data were normally distributed then why all other tests were non-parametric?

- p-value cannot be p=0.000

- if you considered p values under 0.05 as significant, then exact p value 0.05 is not significant

Author Response

The authors improved the significantly the manuscript but there is still some issues with data presentation and statistics. You wrote that some data were normally distributed and therefore the pearson correlation test was used but you presented all data as medians? If some data were normally distributed then why all other tests were non-parametric?

In Section 5.11 Statistical Analysis of the revised version of the manuscript we tried to explain why nonparametric tests were performed also if the hypothesis of normality had been accepted for some data and, consequently, in Tables 1 and 2 the three quartiles are reported. However, now in the box plots of Figure 1 also the mean values are represented.

- p-value cannot be p=0.000

p-value was reported equal to 0.000 owing to digit rounding. In the revised version, following your remark, instead of “0.000” we reported “<0.001”.

 - if you considered p values under 0.05 as significant, then exact p value 0.05 is not significant.
We agree with you. Following your comment, we reported ns (not significant)

Round 3

Reviewer 2 Report

I have no further comment